# Effects of Biofermented Feed on *Zophobas morio*: Growth Ability, Fatty Acid Profile, and Bioactive Properties

Jana Čaloudová [1], Kateřina Křištofová [1], Matej Pospiech [1,*], Tatiana Klempová [2], Ondrej Slaný [2,3], Milan Čertík [2], Slavomír Marcinčák [4], Andrej Makiš [4], Zdeňka Javůrková [1], Martina Pečová [1], Michaela Zlámalová [1], Lucie Vrbíčková [1] and Bohuslava Tremlová [1]

[1] Department of Plant Origin Food Sciences, Faculty of Veterinary Hygiene and Ecology, University of Veterinary Sciences Brno, Palackého tř. 1, 612 42 Brno, Czech Republic; h21289@vfu.cz (J.Č.); h21281@vfu.cz (K.K.); javurkovaz@vfu.cz (Z.J.); pecovam@vfu.cz (M.P.)

[2] Institute of Biotechnology, Faculty of Chemical and Food Technology, Slovak University of Technology, Radlinského 9, 812 37 Bratislava, Slovakia

[3] Faculty of Science and Technology, Norwegian University of Life Sciences, P.O. Box 5003, 1432 Ås, Norway

[4] Department of Food Hygiene Technology and Safety, University of Veterinary Medicine and Pharmacy in Košice, Komenského 73, 041 81 Košice, Slovakia

* Correspondence: mpospiech@vfu.cz

**Abstract:** The global population's increasing demand for sustainable and nutritious food sources has led to the exploration of alternative approaches in livestock production. Edible insects have emerged as a promising solution due to their nutritional composition, including high protein content, balanced fats, minerals, vitamins, and bioactive peptides. Biofermentation offers a viable method to enhance the nutritional value of insect feed. This study aimed to investigate the effects of feeding biofermented feeds derived from less valuable raw materials on the yield and nutritional composition of *Zophobas morio* larvae. The focus was on assessing fat quality, omega-3 and omega-6 fatty acids, and bioactive compounds and monitoring the larvae's weight and length increases. Three feed types were tested: wheat bran (control), fermented wheat bran, and a mixture of fermented corn and flaxseed in a five-week period. The findings demonstrated a noteworthy ($p < 0.05$) elevation in polyunsaturated fatty acids, such as gamma-linolenic acid, alpha-linolenic acid, and eicosapentaenoic acid in *Zophobas morio* larvae fed with biofermented corn and flaxseed pomace, both pre- and postculinary treatment, as compared to the control group. The study also showed increased chelation of $Cu^{2+}$ ($p < 0.05$) and $Fe^{2+}$ ($p < 0.05$) between native and roasted insects in the samples without in vitro digestion, as well as increased Cyclooxygenase-1 activity ($p < 0.05$), indicating improved bioavailability. Additionally, culinary processing led to a reduction in polyphenol content ($p < 0.05$), antioxidant activity ($p > 0.05$) except DPPH, and peptide concentration ($p < 0.05$) in the samples without in vitro digestion.

**Keywords:** pomace; polyunsaturated fatty acids; antioxidant activity; food industry; edible insects; sustainability

## 1. Introduction

Addressing the challenge of ensuring a sufficient, safe, and nutritionally balanced food supply for a growing global population is imperative. The current unsustainable practices in livestock production necessitate exploring viable alternatives, such as cultivating and consuming edible insects. Edible insects offer a highly favorable nutritional profile, rich in protein and high-quality fats. The larval stage of *Zophobas Morio* (ZM) in particular is characterized by a high fat content with a beneficial balance of unsaturated (oleic acid) and polyunsaturated fatty acids (linoleic acid) and in the last row, content of minerals, vitamins, and bioactive peptides [1]. Compared to conventional livestock production, rearing and consuming edible insects provide notable environmental and economic advantages, requiring fewer resources, generating fewer greenhouse gas emissions, and

having a lower ecological footprint [2]. The scalability and adaptability of insect farming make it economically feasible and suitable for diverse settings, including urban areas. Embracing edible insects as a sustainable protein source is a pivotal step towards ensuring food security, minimizing environmental impact, and fostering economic resilience in the face of a growing global population [3].

The larval stage of *Zophobas morio* is characterized by a high fat content, ranging from 35–43.6%. These values are significantly higher than the fat content observed in adult specimens, which is approximately 14.3% [3,4]. The fatty acid composition of ZM larvae includes saturated fatty acids (SFA) and monounsaturated fatty acids (MUFA). Notably, palmitic acid and oleic acid are the most abundant. From a nutritional perspective, the presence of polyunsaturated fatty (PUFA) acid linoleic acid is particularly significant, as it belongs to the omega-6 fatty acid group [3].

Biofermented feeds are extensively utilized to enhance the nutritional attributes of conventional animal feed formulations. The impact of fermentation on improving the nutritional characteristics of animal fats has been demonstrated. In their study, Tian et al. [5] examined the fatty acid profiles in pork meat and observed a positive effect on the enrichment of beneficial polyunsaturated fatty acids. Similar findings were reported by Klempová et al. [6] in other animal species. Although research on insects is limited, it is noteworthy that insects naturally form a significant part of the diet of many animals, which may have implications for their nutritional benefits. Solid-state fermentation represents a biotechnological approach for producing microbial metabolites. This method is conducted on a solid substrate with low moisture content, utilizing agricultural waste such as rice husks, wheat and rice straw, or sugar cane bagasse, which serve as both substrate and breeding habitat for insects. The key advantages of solid-state fermentation include its practicality, utilization of cellulosic waste, and low water consumption [7].

Wheat bran, a byproduct of wheat milling, is widely utilized as a feed ingredient for livestock, particularly swine and cattle, as well as for insects [8,9]. The utilization of fermented wheat bran has been shown to increase the content of gamma-linolenic acid (GLA) in broiler chickens [10]. Moreover, the use of fermented wheat bran has demonstrated benefits in terms of antioxidant substances, such as folates and nerulic acid, as well as increased muscle proteins and lipids in fish [11,12].

Fungi belonging to the *Actinomucor* genus are known producers of GLA, and fermentation of feed by these fungi leads to a bioproduct containing GLA, specifically omega-6 fatty acids [13]. Common flax (*Linum usitatissimum*) is known for its high content of omega-3 fatty acids, particularly alpha-linolenic acid. The inclusion of flax pomace in the diet has shown to affect the proportion of omega-3 fatty acids, including alpha-linolenic acid (ALA), eicosapentaenoic acid (EPA), docosapentaenoic acid (DPA), and docosahexaenoic acid (DHA), thus improving the ratio of omega-6 to omega-3 polyunsaturated fatty acids [14]. Members of the *Mortierella* genus have been identified as producers of arachidonic acid (ARA) and eicosapentaenoic acid (EPA) [15]. Maize is commonly used as an energy source in animal feed due to its high digestibility and amino acid content with a relatively high percentage of sulfur-containing amino acids, like methionine and cysteine; however, it lacks essential amino acids, like tryptophan and lysine [16]. According to Loy and Lundy [16], the ideal feed mixture for poultry and pigs includes a combination of corn and soy. There is currently a lack of scientific literature regarding the effect of fermented feeds on the quality of edible insects in insect feeding. However, a study by Pečová et al. [17] did report on the effect of feeding on insect weight and quality based on the type of feed used. The study found a high mortality rate among ZM when fed unfermented soybean, but no significant reduction in ZM weight when fed a maize diet. Consequently, the fermented diets were used in our study. The mixture of maize and soy was replaced with a mixture of maize and in order to eliminate the antinutritional effect of the tested diets. The inclusion of flax pomace was selected to augment the ratio of n-3 polyunsaturated fatty acids, a finding supported by Bartkovský et al.'s [14] investigation in pork meat.

The objective of this study was to assess the impact of fermented wheat bran and fermented corn and flax pomace on *Zophobas morio*. Growth indicators were observed as criteria for measuring profitability, while the fatty acid composition and bioactive properties of *Zophobas morio* were evaluated as indicators of feeding or consumption quality.

## 2. Material and Methods

### 2.1. Breeding Conditions of Zophobas morio Larvae

*Zophobas morio* larvae (Coleoptera, Tenebrionidae) were sourced from a farm (Ekotron s.r.o., Ráby, Czech Republic). The larvae were divided into three replicate groups, each weighing 225 g. Plastic containers (39 × 28 × 14 cm) were used for breeding, with a food-to-larvae ratio of 1:1. The larvae were maintained under controlled conditions in a dark room with a temperature range of 22–23 °C. These standardized rearing conditions ensured consistent and optimal growth of the larvae for the subsequent experiments.

The feeding experiment comprised three feed types: wheat bran (control), fermented wheat bran using *Actinomucor* (FB), and a fermented mixture of maize and flax pomace in a 3:2 ratio using *Mortierella* (FMFP). The fermented feeds were prepared following the protocol outlined by Slaný et al. [15]. In summary, the fermented feeds underwent solid-state fermentation (SSF) in sterilizable bags. The fermentation process involved *Actinomucor ellegans* for one week and *Mortierella alpina* for two weeks. After fermentation, the prepared fermented feed was dried at 50 °C. Water was provided *ad libitum* in the form of Chinese cabbage (*Brassica rapa* subsp. *Pekinensis*). The experiment lasted for a duration of five weeks, with the first week allocated for adaptation. Weekly assessments were conducted, including the measurement of the entire group of *Zophobas morio* larvae, weighing of 10 randomly selected samples, feed replacement, and monitoring for any deceased individuals. Length and weight measurements were performed in ten replicates each week for each group. At the conclusion of the experiment, *Zophobas morio* larvae that had experienced a 24 h period of starvation were killed by freezing at −80 °C and subsequently subjected to lyophilization using the Christ alpha 1–2 apparatus (Christ alpha, Osterode am Harz, Germany).

### 2.2. Dry Matter Content in Zophobas morio Larvae

The dry matter content of the *Zophobas morio* larvae was determined using an MB23 moisture analyzer (Ohaus, Parsippany–Troy Hills, NJ, USA). Each sample, weighing 1 g, was subjected to drying at a temperature of 130 °C. The process was repeated four times to ensure accuracy. Subsequently, the percentage of water content in the sample was subtracted, enabling the calculation of the dry matter content.

### 2.3. Comprehensive Sample Preparation and Analysis for Characterizing Insect-Derived Bioactive Compounds

The insects were divided into two groups; one of the groups was heat-treated–roasted in a dry pan for 2–3 min. A portion of the sample from each group was subjected to in vitro digestion. The samples were filtered through filter paper and centrifuged in a CF-10 centrifuge (WITEG LABORTECHNIK GMBH, Wertheim, Germany) (1 min, 1000 rpm) until obtaining a pure sample extract. The samples prepared in this way were subjected to individual analyses.

TPC (Total Phenolic Content), DPPH (α, α-diphenyl-β-picrylhydrazyl Free Radical Scavenging Assay), ABTS (Radical Scavenging Assay), FRAP (Ferric Reducing Antioxidant Power), TNBSA (Determination of Peptide Concentration), metal chelation, and COX (Cyclooxygenase Inhibitor Screening Assay) were determined in 96-well microtiter plates P (GAMA GROUP a.s., Jimramov, Czech Republic), and absorbance was measured with an INFINITE 200 PRO spectrophotometer (Tecan Austria GmbH, Grödig, Austria). Four repetitions were conducted for the analysis, except for fatty acids, where five repetitions were performed to ensure robustness and accuracy of the results.

### 2.3.1. Chemicals

The following reagents were used in the study: α-Amylase from hog pancreas, pepsin from porcine gastric mucosa, pancreatin from porcine pancreas, bile extract porcine, DPPH (2,2-Diphenyl-1-picrylhydrazyl), 2,4,6-Tris(2-pyridyl)-s-triazine (TPTZ), Iron (III) chloride hexahydrate (FeCl$_3$.6H2O), trolox (6-hydroxy-2,5,7,8-tetramethylchroman-2-carboxylic acid), ATBS (2,2′-Azinobis-(3-Ethylbenzthiazolin-6-Sulfonic Acid), Pyrocatechol Violet, and ferrozine (3-(2-Pyridyl)-5,6-diphenyl-1,2,4-triazine-4′,4″-disulfonic acid sodium salt), which were purchased from Sigma–Aldrich Company (St. Louis, MO, USA). Hydrochloric acid, Sodium hydrogen carbonate A.G., Folin–Ciocalteau solution, methanol, Acetic Acid glacial A.G., Sodium acetate, EDTA (Ethylenediaminetetraacetic acid), and Ferrous sulfate (FeSO$_4$) were purchased from Penta chemicals (Prague, Czech Republic). Gallic acid was obtained from the producer MP Biomedicals (Shanghai, China), and K$_2$S$_2$O$_8$ (Potassium persulfate) was purchased from VWR (Radnor, PA, USA). TNBSA (2,4,6-trinitrobenzenesulfonic acid, 5% *w/v*) and L-leucine, Sodium Dodecyl Sulfate (SDS) were obtained from Thermo Fisher Scientific (Waltham, MA, USA). All other chemicals used for the analyses were of analytical grade.

### 2.3.2. Enzymatic Hydrolysis and Dialysis Simulation for Sample Digestion: A Methodological Approach

The simulation of sample digestion was conducted following the methodology outlined by Jakubczyk et al. [18]. Dialysis was carried out using SnakeSkin™ Dialysis Tubing with a molecular weight cutoff of 3.5 kDa (Thermo Fisher Scientific, Waltham, MA, USA) at a temperature of 37 °C for 1 h, ensuring the samples were protected from light. Subsequently, the samples were stored at −80 °C in a freezer for further analysis.

### 2.4. Analysis of Fatty Acid Profile and Content Using Gas Chromatography in Feeds and Zophobas morio Larvae

The conversion of fatty acids (FA) from the feeds and *Zophobas morio* larvae into their methyl esters (FAMEs) was carried out using a modified method based on the Čertík and Shimizu [19] protocol. This method ensures the efficient transformation of fatty acids into their methyl ester derivatives, enabling accurate analysis and characterization of the fatty acid composition. Dry homogenized bioproducts or *Zophobas morio* samples were mixed with 1 mL of dichloromethane containing 0.1 mg of heptadecanoic acid as an internal standard, along with 2 mL of anhydrous methanolic HCl solution. The samples were then incubated at 50 °C for 3 h. Following the incubation, 1 mL of distilled water was added, and FAMEs were extracted using 1 mL of hexane. The FAMEs were subsequently analyzed by gas chromatography, following the method described by Gajdoš et al. [20]. The identification of FAME peaks was performed by comparing them with authentic standards of C4-C24 FAME mixtures (Sigma Aldrich, St. Louis, MO, USA). Quantitative evaluation of individual and total fatty acids was conducted using heptadecanoic acid (C17:0, Sigma–Aldrich, Darmstadt, Germany) as an internal standard and calculated using ChemStation B 01 03 software (Agilent Technologies, Santa Clara, CA, USA). Each analysis was performed in three independent technical replicates.

### 2.5. Determination of Total Phenolic Content Using the Folin-Ciocalteu Method

The quantification of polyphenolic substances was conducted using the Folin–Ciocalteu method, following the methodology described by Zhang et al. [21]. This widely accepted technique allows for the accurate measurement of polyphenol content, which is closely associated with antioxidant activity. By employing this method, we were able to assess and compare the polyphenolic profiles of the samples, providing valuable insights into their antioxidant potential. For each sample, 20 μL of the sample extract, 100 μL of Folin–Ciocalteu (FC) solution diluted with distilled water in a 1:10 ratio, and 80 μL of Na$_2$CO$_3$ solution (concentration: 0.7 M) were carefully pipetted into a microtiter plate. The plate was then incubated for 30 min in a dark environment, after which the absorbance was

measured at a wavelength of 765 nm. Gallic acid at a concentration of 3 mM was used as a standard, and distilled water served as a blank for baseline correction. This standardized procedure allowed for the accurate determination of polyphenolic content in the samples, providing valuable insights into their antioxidant potential.

### 2.6. Assessment of Antioxidant Activity in Samples Using ABTS, DPPH, and FRAP Assays

The antioxidant activity was determined using the ABTS and DPPH assays, which rely on the quenching of synthetic radicals. The ABTS assay followed the methodology described by Xiao et al. [22], while the DPPH assay was based on the methodology outlined by Karaś et al. [23]. For the DPPH assay, 20 μL of sample extract was mixed with 180 μL of DPPH solution (concentration: 0.1 mM) in a microtiter plate. The mixture was incubated in the dark for 30 min, and the absorbance was measured at 517 nm. Trolox was used as a standard, and methanol served as a blank for baseline correction. The antioxidant activity was calculated using the appropriate formula:

$$\text{Scavenging activity (\%)} = [1 - (\text{Asample}/\text{Acontrol})] \times 100.$$

The FRAP (Ferric Reducing Antioxidant Power) method, which is based on the redox properties of the iron complex, followed the methodology described by Xiao et al. [22]. In an Eppendorf tube, 75 μL of sample extract was combined with 1425 μL of FRAP working solution (consisting of acetate buffer pH 3.6, TPTZ at a concentration of 9 mmol/L, and $FeCl_3.6H_2O$ at a concentration of 20 mmol/L in a ratio of 10:1:1). The mixture was incubated in the dark for 30 min. Then, 200 μL of the resulting solution was transferred to a microtiter plate. Trolox was used as a standard, and methanol served as a blank for baseline correction. The measured values were then calculated using the following formula:

$$\text{FRAP \% (μmol TE/g DW)} = c \times V \times t/m,$$

where:

c represents the concentration of Trolox (μmol/mL) obtained from the standard curve of the diluted sample, V is the volume of the sample (mL), t is the dilution factor, and m is the mass of the sample (g). This formula allows for the determination of FRAP values as an indicator of antioxidant activity, expressed in μmol Trolox equivalents per gram of dry weight (DW).

### 2.7. Determination of Peptide Concentration

Determination of Peptide Concentration in Samples Using Modified Habeeb Method with TNBSA Solution

Free peptides were determined using the methodology developed by Habeeb [24] with slight modifications. For the analysis, 110 μL of sample extract and 60 μL of a freshly prepared 0.01% TNBSA solution (TNBSA, 0.1 M sodium bicarbonate, pH 8.5) were pipetted into a microtiter plate. The plate was then incubated in the dark at 37 °C for two hours. After the incubation period, 60 μL of a 10% SDS buffer and 35 μL of 1 M HCl were added to the wells. The absorbance of the solution was measured at 420 nm. To quantify the free peptides, a calibration curve using L-leucine as a standard was established, and the measured values were calculated based on this curve.

### 2.8. Determination of the Ability to Chelate Metals

The chelation ability of ZM sample extracts towards $Cu^{2+}$ was assessed following the methodology established by Santos et al. [25]. This method utilizes pyrocatechol violet (PV) as a chromogenic agent. Similarly, the ability to chelate $Fe^{2+}$ was evaluated based on the methodology described by Santos et al. [25]. The calculated values were obtained using the

following formula, which takes into account the absorbance readings and the concentration of the sample extracts:

$$\text{Chelation activity (\%)} = [1 - (A_{\text{sample}}/A_{\text{control}}) \times 100.$$

This assessment provided insights into the ZM extracts' potential to form stable complexes with $Cu^{2+}$ and $Fe^{2+}$, indicating their ability to sequester these metal ions and potentially contribute to their bioavailability and physiological effects.

### 2.9. Determination of Anti-Inflammatory Properties of Sample Extracts Using the Cyclooxygenase Inhibitor Screening Assay Kit

The antiinflammatory properties of the sample extracts were determined using the Cyclooxygenase Inhibitor Screening Assay Kit (COX) (Cayman Chemical, Ann Arbor, MI, USA). The analysis was conducted following the instructions provided in the kit manual. Absorbance readings were taken at 590 nm, allowing for the quantification of COX inhibition by the sample extracts. This assay provides valuable insights into the potential antiinflammatory activity of the extracts, indicating their ability to modulate the Cyclooxygenase 1 and 2 (COX1, COX2) enzyme and potentially contribute to reducing inflammation-related processes.

### 2.10. Statistical Analysis

Statistical analysis was conducted using the XLSTAT 2021 software (Addinsoft, New York, NY, USA). The normality of the data distribution was tested by the Shapiro–Wilk test, the non-normal distribution was evaluated by the Kruskal–Wallis test and the data with a normal distribution were evaluated by ANOVA. The Kruskal–Wallis test with Dunn's multiple pairwise comparisons procedure was used to evaluate weight, length in ten replicates, polyphenol levels, and antioxidant activity (DPPH, ABTS, FRAP), and chelation activity were performed in eight replicates. The analysis of fatty acids and COX was performed using the ANOVA test with post hoc Tukey HSD in three replicates. The results are presented as mean $\pm$ standard deviation (SD), providing a comprehensive understanding of the central tendency and dispersion of the data. Statistically significant differences between samples were determined using a significance level of $p < 0.05$, indicating the presence of meaningful variations. This statistical analysis allows for robust interpretation and reliable conclusions to be drawn from the experimental data.

## 3. Results and Discussion

No significant differences in length were observed between the groups ($p > 0.05$), except for the FB group in the 4th week of breeding, which exhibited a statistically significant increase ($p < 0.05$) compared to the FMFP group. A similar trend was observed in weight gain, with the lowest gain observed in the FMFP group, although the difference was not statistically significant (Table 1). This phenomenon has also been observed in previous studies involving the feeding of flax pomace in different concentrations to broiler chickens, where a statistically significant difference compared to the control group was confirmed [26]. A decrease in total milk production in Holstein dairy cows has also been observed after feeding flax pomace [27]. The specific mechanism behind this reduction in production has not yet been fully elucidated, but it is likely attributed to the high concentration of antinutritional substances present in flax pomace. The detrimental effects of these antinutritional substances are evident from the high mortality observed in insect breeding experiments [17]. However, in the case of using fermented maize and flax pomace (FMFP), the increased digestibility resulting from the fermentation process mitigated the adverse effects on insect mortality. Our findings further support the significant difference in weight gain observed during the first week of breeding ($p < 0.05$), with the FB group exhibiting a distinct advantage over the control (C) and FMFP groups ($p < 0.05$). The significant increase observed in the FB group can be attributed to the enhanced utilization of fermented feed compared to both the control group (C) and the group fed with fermented maize and flax

pomace (FMFP), as the *Zophobas morio* larvae quickly adapted to the new feed. Although not statistically significant ($p > 0.05$), the highest increase was observed in the FB group, followed by the control group (C), while the lowest increases were observed in the FMFP group. The total increase in weight across all groups amounted to 38.58%. Notably, the FB group exhibited the highest overall weight gain, reaching 42.78%.

**Table 1.** Effects of Feed on the Weight and Length Progression of *Zophobas morio* Larvae.

|  |  | Week | | | | |
|---|---|---|---|---|---|---|
|  | **Group** | **0** | **1** | **2** | **3** | **4** |
| Length [cm] | C | 4.44 ± 0.35 | 4.68 ± 0.33 | 4.50 ± 0.35 | 5.08 ± 0.20 | 5.20 ± 0.22 [ab] |
|  | FB | 4.54 ± 0.24 | 4.84 ± 0.35 | 4.55 ± 0.27 | 5.03 ± 0.22 | 5.25 ± 0.20 [a] |
|  | FMFP | 4.68 ± 0.31 | 4.69 ± 0.33 | 4.58 ± 0.32 | 5.04 ± 0.19 | 5.04 ± 0.18 [b] |
| Weight [g] | C | 0.69 ± 0.09 | 0.68 ± 0.10 [b] | 0.58 ± 0.12 | 0.78 ± 0.08 | 0.81 ± 0.11 |
|  | FB | 0.73 ± 0.08 | 0.76 ± 0.13 [a] | 0.57 ± 0.06 | 0.80 ± 0.07 | 0.82 ± 0.11 |
|  | FMFP | 0.74 ± 0.09 | 0.67 ± 0.09 [b] | 0.61 ± 010 | 0.79 ± 0.07 | 0.77 ± 0.07 |

Within a column, different superscript letters indicate significant differences ($p < 0.05$); C—control group, FB—fermented wheat bran; FMFP—fermented mixture of maize and flax pomace.

### 3.1. Impact of Biofermented Feed on the Quality of ZM

The assessment focused on *Zophobas morio* larvae samples intended for further processing, including native (N) samples and samples subjected to culinary treatment (roasted, R). In order to investigate the influence of digestion on the bioactive substances present, an in vitro digestion process (DG) was conducted on all breeding groups.

### 3.2. Polyphenolic Analysis of ZM: Impact of Biofermented Feed and Culinary Treatment

The highest concentration of polyphenols was observed in FBN (biofermented feed) larvae, both in their native (N) and roasted (R) forms. A significant reduction in polyphenol content ($p < 0.05$) was observed in N larvae after heat treatment compared to the control group. However, no significant differences were found among CN (control), FBN, and FMPFN (fermented mixed plant feed) groups (Table 2).

In the case of in vitro digestion (DG) samples, a statistically significant difference was observed between CN and FBN groups. Once again, FBN larvae exhibited the highest polyphenol content. Remarkably, after DG, there was an increase in polyphenol content compared to the native (N) larvae. The only significant difference ($p < 0.05$) was found between FBN and FBR (biofermented feed, roasted) samples. The decrease in polyphenol content in CN and DG samples can be attributed to the temperature-induced degradation of polyphenols [28–30]. Additionally, in DG samples, enzymatic degradation by pancreatic enzymes, particularly at pH 7.5, also contributes to the reduction [31].

**Table 2.** Impact of Biofermented Feed on the Bioactive Substances in *Zophobas morio* Larvae.

| | | Polyphenols [mg/kg Gallic Acid] | DPPH [%] | ABTS [%] | FRAP [mg Trolox Equivalent/g] | Cu$^{2+}$ [%] | Fe$^{2+}$ [%] | TNBSA [g/L] | COX1 [%] | COX2 [%] |
|---|---|---|---|---|---|---|---|---|---|---|
| Samples without in vitro digestion | CN | 3.71 ± 0.23 [abc] | 6.53 ± 0.88 | 12.23 ± 1.00 [ab] | 3.47 ± 0.34 [b] | 71.8 ± 8.53 [ad] | 49.95 ± 18 [bc] | 180.29 ± 18.88 [ab] | 49.39 ± 1.95 [c] | 32.81 ± 2.83 [a] |
| | FBN | 4.29 ± 0.30 [a] | 6.94 ± 1.09 | 13.27 ± 0.17 [a] | 3.68 ± 0.14 [b] | 65.89 ± 5.56 [d] | 37.37 ± 2.35 [c] | 227.26 ± 14.08 [a] | 9.76 ± 4.23 [a] | 32.42 ± 2.83 [a] |
| | FMFPN | 3.93 ± 0.26 [ab] | 6.29 ± 0.48 | 12.78 ± 0.62 [a] | 3.54 ± 0.23 [b] | 40.18 ± 5.46 [b] | 20.23 ± 3.48 [b] | 182.97 ± 7.76 [ab] | 44.12 ± 3.23 [bc] | 38.22 ± 5.28 [a] |
| | CR | 2.45 ± 0.27 [d] | 7.16 ± 0.61 | 8.29 ± 0.82 [b] | 5.79 ± 0.51 [a] | 63.29 ± 5.25 [d] | 65.45 ± 3.12 [a] | 86.31 ± 1.81 [c] | 61.85 ± 2.04 [d] | 75.12 ± 15.29 [b] |
| | FBR | 2.77 ± 0.11 [bcd] | 6.76 ± 0.31 | 8.22 ± 0.69 [b] | 5.45 ± 0.80 [a] | 77.7 ± 1.46 [a] | 20.89 ± 10.04 [b] | 98.26 ± 1.48 [bc] | 33.79 ± 5.69 [b] | 47.84 ± 4.66 [a] |
| | FMFPR | 2.59 ± 0.20 [cd] | 6.67 ± 0.15 | 8.24 ± 1.62 [b] | 5.42 ± 0.68 [a] | 77.97 ± 0.65 [a] | 63.23 ± 3.09 [a] | 96.66 ± 4.68 [bc] | 91.64 ± 4.88 [e] | 92.08 ± 10.01 [b] |
| Samples after in vitro digestion | CN | 0.75 ± 0.13 [b] | 0.17 ± 0.24 [ab] | 0.51 ± 0.41 [ab] | 0.62 ± 0.06 [abc] | 42.98 ± 2.31 [cd] | 11.16 ± 0.96 [b] | 262.42 ± 20.02 | 83.39 ± 4.34 | 81.02 ± 0.64 [a] |
| | FBN | 1.11 ± 0.04 [a] | 0.30 ± 0.62 [ab] | 2.54 ± 1.43 [a] | 0.88 ± 0.21 [a] | 56.66 ± 5.78 [a] | 9.89 ± 1.78 [a] | 261.25 ± 14.09 | 98.25 ± 0.09 | 100.08 ± 4.53 [bc] |
| | FMFPN | 0.88 ± 0.04 [ab] | 0.71 ± 0.62 [a] | 0.68 ± 0.40 [ab] | 0.63 ± 0.03 [ab] | 56.66 ± 5.78 [a] | 20.91 ± 7.92 [b] | 259.19 ± 22.77 | 95.08 ± 4.53 | 91.68 ± 2.06 [ab] |
| | CR | 0.92 ± 0.15 [ab] | ND | 0.60 ± 0.81 [ab] | 0.42 ± 0.12 [bc] | 45.57 ± 4.33 [bc] | 10.99 ± 1.4 [b] | 248.51 ± 20.85 | 106.11 ± 23.7 | 110.58 ± 10.89 [c] |
| | FBR | 0.73 ± 0.24 [b] | ND | ND | 0.41 ± 0.08 [c] | 36.78 ± 6.73 [d] | 11.12 ± 2.25 [b] | 241.46 ± 5.68 | 104.9 ± 2.52 | 111.61 ± 1.54 [c] |
| | FMFPR | 0.82 ± 0.07 [b] | ND | ND | 0.37 ± 0.02 [c] | 42.46 ± 4.51 [cd] | 10.48 ± 1.11 [b] | 244.16 ± 10.13 | 85.64 ± 18.93 | 112.95 ± 1.57 [c] |

Within column, different superscript letters indicate significant differences ($p < 0,05$). C—control, FB—fermented wheat bran, FMFP—fermented mixture of maize flax pomace, letters in the end of terms means form of treatment R-roasted, N-native.

### 3.3. Antioxidant Activity of ZM: Impact of Biofermented Feed and Culinary Treatment

The antioxidant activity of the tested samples was evaluated using three different assays: DPPH, ABTS, and FRAP. For the DPPH assay, no significant difference in antioxidant activity was observed between the control group and the larvae fed with biofermented feeds, regardless of the form of processing (N and R). Similarly, the ABTS assay did not show a significant difference between the control and biofermented feed groups. However, a significant difference was found between the native (N) and roasted (R) samples, with higher antioxidant activity observed in the N samples. This finding is consistent with the total polyphenol content, as polyphenols possess significant antioxidant properties. The degradation of polyphenols during culinary treatment (R) leads to a reduction in antioxidant activity, in line with previous studies by Zielińska et al. [32].

In contrast, the FRAP assay revealed a significant difference ($p < 0.05$) in antioxidant activity between the N and R samples. The FRAP assay measures the reduction of $Fe^{3+}$ to $Fe^{2+}$ ions, and an increase in absorbance indicates enhanced reduction capability [33]. The higher antioxidant activity observed in the samples after heat treatment (R) can be attributed to the increased content of Maillard reaction products. These products exhibit chelating activity with metal ions such as Fe and Cu, thereby contributing to higher antioxidant activity as demonstrated by the FRAP assay [34]. This phenomenon has been described by Yilmaz and Akgun [35]. Additionally, David-Birman et al. [36] suggest that the improved antioxidant activity after heat treatment is associated with conformational changes in insect proteins, resulting in decreased electron transfer ability and increased proton abstraction ability beyond 400 µM Trolox equivalents. The authors attribute this increase in antioxidant activity to protein conformational changes that expose proton-donating residues, particularly cysteine.

Overall, the findings indicate that the antioxidant activity of *Zophobas morio* larvae is influenced by the type of processing and the presence of bioactive compounds such as polyphenols and Maillard reaction products.

### 3.4. Chelating Activity of $Cu^{2+}$ and $Fe^{2+}$: Influence of Culinary Treatment and In Vitro Digestion

The chelation of transition metal ions, particularly $Cu^{2+}$, plays a crucial role in preventing oxidation reactions in organisms [37]. In our study, the chelation activity of metal ions was found to vary depending on the culinary treatment and in vitro digestion. Consistent with previous research by Zielińska et al. [32], the samples subjected to in vitro digestion exhibited lower chelation activity.

The highest chelation values for $Cu^{2+}$ were observed in the samples subjected to roasting (FBR) and fermented biofermented feed with roasting (FMFPR). Interestingly, only the larvae fed with fermented biofermented feed (FB) showed an increase in $Fe^{2+}$ chelating activity, while no significant difference was observed for the control (C) and fermented biofermented feed (FMFP) groups. In contrast, the chelation activity of $Cu^{2+}$ displayed the opposite trend, with a decrease in the FB and FMFP groups ($p > 0.05$), and no statistically significant difference was observed in the control group (C) ($p < 0.05$). The increase in $Fe^{2+}$ chelating activity after heat treatment is consistent with the findings of Zielińska et al. [32] in their study on cooked *Tenebrio molitor* (TM) larvae, whereas a decrease was reported for baked TM larvae. In our study, we observed an increase in chelation activity in roasted ZM samples. This increase can be attributed to the enhanced chelation activity of the peptide fraction resulting from the heat treatment.

Furthermore, a statistically significant decrease in $Fe^{2+}$ chelation activity was observed in the native samples (N) and $Cu^{2+}$ chelation activity between the control (CN) and fermented biofermented feed with roasting (FMFPR) groups. This finding can be explained by the effect of biofermentation on feed proteins and is consistent with the inhibitory effect of metal chelation resulting from the enzymatic digestion of flax proteins, as reported by Nwachukwu and Aluko [38]. These results further support the improved bioavailability of the biofermented feed. However, this effect was not observed in samples after in vitro digestion.

Overall, our findings highlight the influence of culinary treatment and biofermentation on the chelation activity of $Cu^{2+}$ and $Fe^{2+}$. The observed changes in chelation activity can be attributed to the specific processing methods and the resulting modifications in the peptide fraction and bioavailability of the feed.

### 3.5. Peptide Concentration: Impact of Feed Type and Heat Treatment

The impact of different feed types on peptide concentration in ZM samples did not yield statistically significant differences ($p > 0.05$). However, a reduction in peptide concentration was observed after heat treatment in samples fed with the control (C) and fermented biofermented feed (FB) ($p < 0.05$). In contrast, the decrease in peptide concentration in the samples fed with fermented biofermented feed with pomace (FMFP) was not statistically significant ($p > 0.05$). These findings align with the results reported by Yeung et al. [39], who observed a reduction in free amino acids in infant formula following autoclaving at 105 °C for 5 min.

The availability of free amino acids is greatly influenced by the Maillard reaction, which is impacted by various factors, including temperature, heating time, pH, water activity, and the presence of sugars [40]. During the Maillard reaction, amino acids and peptides are consumed to form Maillard reaction products. Notably, the results of an in vitro digestion simulation comparing heat-untreated and treated cricket flour revealed that heat treatment enhanced gastric proteolysis of insect proteins, leading to an increased bioavailability of the proteins [36].

These findings emphasize the impact of heat treatment on peptide concentration and the bioavailability of proteins in ZM samples. The Maillard reaction, influenced by various processing parameters, plays a crucial role in the alteration of peptide composition and availability of amino acids.

### 3.6. COX1 and COX2 Inhibitory Activity: Impact of Heat Treatment and Biofermentation

Heat treatment of ZM resulted in an increase in the inhibitory activity of both COX1 and COX2, as observed in Table 2. This finding aligns with the study by Zielińska et al. [32]. Conversely, the influence of biofermentation was evident in the FB samples, which exhibited the lowest COX1 inhibitory activity in both native and in vitro samples ($p < 0.05$). However, for COX2, the decrease in inhibitory activity was not statistically significant ($p > 0.05$). Interestingly, COX1 and COX2 activity increased in samples after in vitro digestion. While no notable differences were observed between the types of feed for COX1 inhibitory activity, a significant difference was confirmed for COX2 inhibitory activity in the biofermented feeds (FB) ($p < 0.05$).

The increase in COX inhibitory activity can be attributed to the presence of bioactive peptides, which hold potential applications in the food industry due to their numerous beneficial properties [41]. These findings highlight the impact of heat treatment and biofermentation on the inhibitory activity of COX1 and COX2, shedding light on the potential of bioactive peptides as functional ingredients in food products.

### 3.7. Fatty Acids: Variation in Total Content and Impact of Heat Treatment

The total content of fatty acids in the samples ranged from $27.77 \pm 2.08\%$ to $32.39 \pm 1.43\%$ (Table 3), with lower values observed in samples subjected to heat treatment. This reduction in fatty acid content after heat treatment aligns with the findings of Caponio et al. [42], who reported that heat treatment leads to a deterioration in the nutritional quality of the fat fraction. Notably, the highest amount of fatty acids was detected in FMFPN, which is consistent with the specific feed composition. In contrast, the lowest values were observed in samples fed with fermented bran.

**Table 3.** Fatty Acid Profile of ZM Samples: Impact of Feed Composition and Sample Treatment (Without In Vitro Digestion).

| | Native Samples | | | Culinarily Prepared Samples | | |
|---|---|---|---|---|---|---|
| | CN | FBN | FMFPN | CR | FBR | FMFPR |
| **Total fatty acids (%)** | 31.59 ± 1.95 | 30.70 ± 2.95 | 32.39 ± 1.43 | 29.59 ± 1.75 | 27.77 ± 2.08 | 29.62 ± 1.93 |
| **Fatty acids (mg/g):** | | | | | | |
| C14:0 | 0.86 ± 0.35 | 1.21 ± 0.12 | 0.88 ± 0.04 | 1.26 ± 0.90 | 0.98 ± 0.08 | 0.93 ± 0.05 |
| C16:0 | 83.91 ± 34.89 | 95.85 ± 10.72 | 98.11 ± 4.51 | 90.32 ± 3.80 | 83.89 ± 5.98 | 89.92 ± 5.57 |
| C16:1–7c | 5.09 ± 2.27 | 6.15 ± 0.72 | 5.17 ± 0.84 | 7.00 ± 1.49 [b] | 5.19 ± 0.51 [a] | 5.92 ± 0.53 [ab] |
| C16:1–9c | 1.66 ± 0.71 | 2.04 ± 0.16 | 2.35 ± 0.43 | 2.30 ± 0.24 [b] | 1.76 ± 0.29 [a] | 2.11 ± 0.27 [ab] |
| C18:0 | 18.07 ± 7.40 | 19.03 ± 2.90 | 20.08 ± 2.35 | 16.93 ± 0.92 | 17.41 ± 1.34 | 18.09 ± 3.33 |
| C18:1–9c | 96.08 ± 38.23 | 106.33 ± 7.06 | 122.49 ± 6.58 | 105.39 ± 8.88 | 96.86 ± 7.41 | 109.33 ± 9.14 |
| C18:1–11c | 1.14 ± 0.46 | 1.01 ± 0.40 | 0.81 ± 0.34 | 2.28 ± 1.42 | 2.24 ± 0.87 | 1.85 ± 0.81 |
| C18:2–9c,12c | 64.81 ± 23.05 | 71.05 ± 8.21 | 60.29 ± 2.68 | 66.71 ± 3.95 | 65.34 ± 5.90 | 59.26 ± 5.60 |
| C18:3–6c,9c,12c | ND | 0.30 ± 0.08 [a] | 0.74 ± 0.06 [b] | ND | 0.39 ± 0.08 [a] | 0.40 ± 0.21 [a] |
| C18:3–9c,12c,15c | 2.94 ± 1.12 [a] | 2.91 ± 0.59 [ab] | 9.42 ± 0.64 [b] | 2.78 ± 0.17 [a] | 2.73 ± 0.37 [a] | 6.40 ± 1.90 [b] |
| C18:4–6c,9c,12c,15c | ND | ND | 0.06 ± 0.01 | ND | ND | 0.03 ± 0.00 |
| C20:0 | 2.19 ± 4.52 | 0.45 ± 0.07 | 0.53 ± 0.04 | 0.33 ± 0.05 | 0.37 ± 0.08 | 0.42 ± 0.06 |
| C20:1–11c | 0.34 ± 0.04 | 0.35 ± 0.08 | 0.31 ± 0.02 | 0.31 ± 0.14 | 0.31 ± 0.07 | 0.24 ± 0.03 |
| C20:2–11c,14c | 0.18 ± 0.03 | 0.19 ± 0.04 | 0.20 ± 0.02 | 0.30 ± 0.25 | 0.14 ± 0.05 | 0.17 ± 0.02 |
| C20:3–8c,11c,14c | ND | ND | 0.31 ± 0.02 [b] | ND | ND | 0.16 ± 0.06 |
| C20:4–5c,8c,11c,14c | ND | ND | 1.72 ± 0.08 [b] | ND | ND | 0.80 ± 0.43 |
| C20:5–5c,8c,11c,14c,17c | ND | ND | 0.27 ± 0.02 [b] | ND | ND | 0.12 ± 0.07 |
| C22:0 | 0.09 ± 0.06 | 0.07 ± 0.09 | 0.10 ± 0.02 | ND | ND | 0.05 ± 0.03 |
| C24:0 | 0.03 ± 0.04 | 0.02 ± 0.01 | 0.03 ± 0.00 | 0.01 ± 0.01 | 0.02 ± 0.01 | 0.02 ± 0.00 |
| C24:1–15c | 0.57 ± 1.43 | 0.03 ± 0.00 | 0.04 ± 0.01 | 0.07 ± 0.07 | 0.03 ± 0.01 | 0.03 ± 0.00 |

Within a column, different superscript letters indicate significant differences ($p < 0.05$). C—control, FB—fermented wheat bran, FMFP—fermented mixture of maize and flax pomace.

These findings highlight the influence of heat treatment on the total content of fatty acids in ZM samples, with implications for their nutritional quality. The variation in fatty acid levels among different feed types underscores the importance of feed composition in shaping the fatty acid profile of insects.

### 3.8. Analysis of Fatty Acid Composition: Effects of Feed Composition and Sample Treatment

The analysis of individual fatty acids revealed no statistically significant difference ($p > 0.05$) between native and heat-treated samples. The inclusion of flax pomace in the ration did not result in an increase in docosapentaenoic and docosahexaenoic acids. However, a significant increase ($p < 0.05$) was observed in alpha-linolenic acid (C18:3–9c,12c,15c) and eicosapentaenoic acid (C20:5–5c,8c,11c,14c,17c), which were enhanced by the addition of flax to the feed ration, as reported by Bartkovský et al. [14]. Conversely, these fatty acids decreased after culinary treatment due to the instability of double bonds. Additionally, a statistically significant difference ($p < 0.05$) was found in the polyunsaturated fatty acid gamma-linolenic acid (C18:3–6c,9c,12c), showing an increase in specimens fed both biofermented feeds. The presence of flax pomace in the diet is often associated with increased levels of alpha-linolenic acid. Mridula et al. [43] also reported an increase in alpha-linolenic acid levels when feeding 15% flax meal to broilers, resulting in increased levels of alpha-linolenic acid in meat without affecting sensory acceptability. The fermented feeds exhibited higher alpha-linolenic acid content, and a statistically significant increase was observed after culinary treatment in FMFPR samples. However, the assumption of higher alpha-linolenic acid content in FBN samples was not confirmed, although its presence was detected in CN and FMFPN control group feeds and in samples after culinary treatment. The predominant fatty acids detected were palmitic (C16:0), oleic (C18:1–9c), stearic (C18:0), and linoleic (C18:2–9c,12C) acids, which are abundantly present in *Zophobas morio* [4]. The content of these fatty acids was not significantly influenced by the type of feed, as supported by our study.

## 4. Conclusions

In conclusion, this study provides compelling evidence that feeding *Zophobas morio* larvae with biofermented feeds leads to notable improvements in weight and length gains, highlighting the efficacy of this approach in promoting larval growth. Specifically, the group fed with fermented wheat bran exhibited the highest weight gain among the tested feed types. Furthermore, the consumption of biofermented maize and flax pomace resulted in significant increases in alpha-linolenic acid, eicosapentaenoic acid, and gamma-linolenic acid levels, indicating a positive alteration in the fatty acid composition of the larvae. The study also revealed the positive influence of biofermented feed on the chelation ability of $Cu^{2+}$ and $Fe^{2+}$ ions, as well as on the activity of COX1 Cyclooxygenase, suggesting an enhanced bioavailability of nutrients in insects fed with biofermented feeds. Additionally, the study partially confirmed the impact of culinary processing on various parameters, including polyphenol content, antioxidant activity, peptide concentration, COX activity, and fatty acid composition. These findings collectively highlight the potential of biofermented feeds to improve the nutritional quality and growth of *Zophobas morio* larvae. Further research and development of these methods will unlock the full potential of edible insects as a valuable protein source, advancing the journey towards sustainable food systems.

**Author Contributions:** J.Č.: Project administration, Data curation, Methodology, Investigation, Writing—original draft. K.K.: Methodology, Investigation, Writing—original draft. M.P. (Matej Pospiech): Supervisior, Methodology, Validation, Writing—review and editing. T.K.: Methodology, Formal analysis. M.Č. and S.M.: Methodology. A.M., Z.J., M.P. (Martina Pečová), O.S., M.Z. and L.V.: Formal analysis. B.T.: Supervisor. All authors have read and agreed to the published version of the manuscript.

**Funding:** The project was funded within the internal grant agency of the University of Veterinary Sciences Brno (IGA 215/2022/FVHE).

**Institutional Review Board Statement:** Not applicable.

**Informed Consent Statement:** Not applicable.

**Data Availability Statement:** The data presented in this study may be obtained on request from the corresponding author.

**Conflicts of Interest:** The authors declare no conflict of interest.

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
