# Peer review of "Effects of Biofermented Feed on Zophobas morio: Growth Ability, Fatty Acid Profile, and Bioactive Properties"

_sustainability, doi:10.3390/su15129709_

Round 1

Reviewer 1 Report

The introduction is not clear, and most of the sentences are not linked together to understand the hypothesis of this study!! you concentrated o the fat content of the fermented feeds but no mentions were made of the chemical composition or the nutritional value of the fermented feeds (e.g., crude protein, fiber). Another issue is that the high content of ash in wheat brans limits their nutritional value for poultry and pig diets, so how the fermented wheat brans may overcome this problem? The part of martial and methods is not clear and needs to be reorganized again, there is no relation between the introduction and the experimental parameters done in this study.

Line 17, it seems that the sentence "on the other hand..." has no relation with the previous sentence.

In the abstract, please insert the P significance values (e.g., P<0.0X) for the obtained results.

The three experimental treatments are not clear in the abstract.

Line 20, this sentence needs English editing.

Line 21, FA has to be inserted in full name in its first mention, and please check through the manuscript that all abbreviations were mentioned in full names in their first mentions.

Line 24, an increase of weight gain for what?

Line 36 needs English editing.

Line 39, in relation to fatty acid profiles in pigs!! do you mean in relation to the fatty acid profiles of pork?

In the introduction, what is the relation of the Larval stage of Zophobas morio with the animal feeds? you have to present evidence of using such larva as a feed source for farm animals, and which animal may get an advantage when consuming this larva? and why?

Line 45, the substrate of what?

Line 49, for which animal?

If the ideal ratio of feed rations in the diet of poultry and pigs is a mixture of corn and soy (in which ratio?), then you feed the larva with a mixture of corn and flax pomace, does it mean you will feed poultry or pigs with this mixture of corn and flax pomace? why do you assume that the fermented mix will serve as animal feed? what was the evidence that you can get the same results when you will use these three feeds for animals?

Line 76, what are these conditions?

Line 86, for which samples?

Line 81, which animals!!

How did you get the three feeds? and how the feeds were fermented. this issue has to be the first part of the martial and methods.

Line 98, the full names are required here.

Total phenolic content was done for what and why?

how many statistical repetitions were done for each parameter in the part of the statistical analysis, what was the experimental unit here? please insert the statistical model you used. you have a time effect in this study why you did not include it in the statistical test? I mean you have to insert the treatment effect, time and time, and treatment interaction effect.

There is no mention of the Fe2 chelate effects in the methods part.

please remove the significance values in the part of the conclusions

The manuscript needs depth English editing

Reviewer 2 Report

The present manuscript is concise and a pleasure to read. It contributes to the knowledge about mass rearing of the, in this regard, scarcely discussed Zophobas species. Whereas T. molitor is extensively studied in this context, there is a lack of knowledge regarding feeding regimes and their consequences on Z. morio. In particular, the intention to improve the digestibility of agricultural by-products by fermentation is a very promising approach, regrading the sustainable use of process by-products as feed for farmed insects. It is recommended to publish this manuscript after minor revisions. 

The quality of english language is adequate. Minor editing is required. 

Reviewer 3 Report

The introduction is not well written and confuses the reader.

Authors should start with a major theme and describe what the issue is and what the purpose of this study is.

In 2.2, there is no mention of any analytical methods for nutrients. Is water a nutrient?

In 2.4, SSF means did authors use INFOGEST? Authors should state so in the analytical method.

But why did autors use Oral Phase SSF? If authors are looking at fatty acid composition toward digestion and absorption, shouldn't authors use the SGF matrix?

2.10 Did authors do ANOVA and Tukey because the fatty acid and COX data were normally distributed? Did the other data not follow a normal distribution? It should be stated that tests of normality and tests of equal variances were performed.

How did authors perform the statistical procedures in Table 1? Multiple comparisons cannot be made with the Kruskal-Wallis test. Why are authors able to make comparisons between groups?

There are so many unacceptable description of the paper just in the beginning.

The manuscript needs to be significantly rewritten, and authors need to have it English proofread service.

Instead of getting proofread for grammars, Authors should get proofread for the way to write scientific paper.

Round 2

Reviewer 1 Report

Line 26, compared to what?

keywords, please write the full name  of the abbreviation "PUFA".

Line 27, are these variations significant if so please insert the P value.

Line 29, "influenced" how?  and insert the significance value.

Line 44, you need a reference after the word "footprint"

Lines 50 and 51 you need suitable references here.

Line 57, what do you mean by current animal feeds??

Line 85, this sentence is not clear.

Line 85, which amino acid? and add a reference.

Why you used flax pomace? may you have more information about it.

Line 92, combination!! but you used individual diets not combinations?

Line 110, ad libitum in italics.

Line 109, Please add more information about the preparation of the fermented feeds.

for the statistical analysis, please insert the number of the number of  statistical repetitions. 

Some sentences are not well understood and need to be clarified.
